# Bioactive Lipid Compounds as Eco-Friendly Agents in the Diets of Broiler Chicks for Sustainable Production and Health Status

**DOI:** 10.3390/vetsci10100612

**Published:** 2023-10-09

**Authors:** Ahmed A. A. Abdel-Wareth, Jayant Lohakare

**Affiliations:** 1Poultry Center, Cooperative Agricultural Research Center, Prairie View A&M University, Prairie View, TX 77446, USA; 2Department of Animal and Poultry Production, Faculty of Agriculture, South Valley University, Qena 83523, Egypt

**Keywords:** broiler chick, blood biochemistry, phytogenic, production, meat quality, sustainability

## Abstract

**Simple Summary:**

Phytogenics provide a variety of biologically active compounds that are advantageous in modern poultry production due to their antibacterial, antioxidative, and digestion-enhancing properties. The current study aimed to determine the effects of the bioactive lipid compounds of oregano and peppermint on the sustainability of meat production and the health of broiler chicks in hot climatic environments. The birds were fed a control diet, oregano bioactive lipid compounds (150 mg/kg), peppermint bioactive lipid compounds (150 mg/kg), and their combination for 35 days. Broilers fed diets containing 150 mg/kg of oregano and peppermint bioactive lipid compounds, either alone or in combination, improved growth performance, health status, and meat quality.

**Abstract:**

Phytogenic compounds can improve feed efficiency, meat quality, and the health status of chickens under hot climatic conditions. The current study investigated the impact of the bioactive lipid compounds of oregano and peppermint and their combination on the sustainability of meat production and the health of broiler chicks in hot climatic conditions. Two hundred and fifty-six one-day-old broiler chicks were distributed into four treatment groups. The birds were fed a control diet, bioactive lipid compounds of oregano (BLCO, 150 mg/kg), bioactive lipid compounds of peppermint (BLCP, 150 mg/kg), or a combination of BLCO and BLCP at 150 mg/kg each for 35 days. Each treatment included 8 replicates, each with 8 birds. The results showed that adding BLCO and BLCP separately or in combination to broiler diets improved body weight, body weight gain, and feed conversion ratio. BLCO, BLCP, or their combination increased the percentages of the dressing and gizzard and lowered the percentage of abdominal fat as compared to the control. Supplementation of BLCO, BLCP, or their combination decreased serum cholesterol, triglycerides, aspartate aminotransferase, alanine transaminase, creatinine, and urea compared to control. BLCO, BLCP, or their combination reduced cook and drip loss in the meat of broilers. In conclusion, birds fed diets containing BLCO and BLCP, either independently or in combination, showed improvements in performance, blood biochemistry, and meat quality in hot climatic conditions.

## 1. Introduction

For sustainable broiler chicken production and superior meat quality, high ambient temperatures must be controlled. According to Lara and Rostagno [1] and Niu et al. [2], one of the most significant environmental stressors that hinders the development of broiler chickens worldwide is heat stress. The heat stress on broilers can result in a variety of negative outcomes, including stunted growth and a decline in the safety and quality of the meat [1]. Heat stress can also cause growth retardation, endotoxemia, systemic and hepatic inflammation, disruption of gut tight junctions, and the generation of reactive oxygen species that promote intestinal permeability [3]. Production of broiler chickens in hot climates continues to face significant challenges in terms of food quality and environmental conditions.

Due to the severe criticism of using antibiotics in poultry diets as growth-enhancing agents [4,5,6]. The utilization of phytogenics such as essential oils, plant extracts, and herbal ingredients as efficient alternatives has been the subject of substantial investigation [7,8,9]. 

Plant bioactive lipid compounds (BLC) are often utilized as phytogenics and are well-known for their antibacterial, potent antioxidant, gastroprotective, appetite-stimulating, and mucus-enhancing effects [10]. Due to their BLC, medicinal plants such as oregano and peppermint may have antioxidant, antimicrobial, and hypocholesteremic effects and enhance the immune system, which may improve poultry performance and meat quality [9,10,11]. Although it is believed that phytogenics’ impacts on the gut microbiota are crucial for the biological effects, the precise mode of action is still unclear [12]. The aromatic perennial herbs oregano and peppermint, grown in Egypt and most other countries, have long been used in medicine [13]. Dietary oregano essential oil supplementation has improved the gastrointestinal secretions, digestion, and health of grill birds [14], which could improve the sustainability of broiler production. 

The practical application of oregano BLC or peppermint BLC in chicken feeding, particularly for broilers, in hot climates is not well documented. It is very challenging to directly compare studies that use various phytogenic products because the effectiveness of those substances also depends on aspects including species, contents, application dose, technique and duration of usage, bird age, as well as stressors from the environment [15]. Additionally, it is still unclear if certain impacts are the consequence of a single component or a synergistic impact produced by a number of components [9,13,16]. Furthermore, there is very little information available regarding the impact of oregano and peppermint BLC on the sustainability of broiler production during heat stress. Henceforth, this study investigated the effectiveness of phytogenics such as oregano BLC and peppermint BLC, either independently or in combination, in enhancing the growth performance, meat quality, and blood biochemicals of broiler chickens under hot climatic conditions.

## 2. Materials and Methods

### 2.1. Design of the Experiment and Nutritional Treatments

Four dietary treatments were applied to a total of 256 one-day-old Ross 308 unsexed broiler chicks. Treatment groups were given either a standard diet or a standard diet with oregano’s bioactive lipid components (BLCO, 150 mg/kg), a standard diet with the bioactive lipid compounds of peppermint (BLCP, 150 mg/kg), or a standard diet with a combination of BLCO and BLCP, each at the 150 mg/kg level. Each treatment had eight replicate pens with eight birds each. The dosages used were selected based on prior individual experiments with increasing levels of peppermint or oregano BLC, and the highest performance was observed at 150 mg/kg [9,13,16]. The experiments lasted for 35 days and consisted of two phases: starter (0–21 days) and grower (22–35 days). The nutritional requirements of Ross 308 Aviagen were used to formulate the experimental diets [17] for broilers (Table 1).

### 2.2. Analysis of Bioactive Lipid Compounds in Peppermint and Oregano

The BLCO and BLCP were hydro-distilled for three hours from dried leaves using a Clevenger-style device (Glassco, Haryana, India) at the National Research Center at the Medicinal and Aromatic Plants Research Department, Egypt. The extract was examined using a gas chromatograph (Waltham, MA, USA: Thermo Fisher Scientific Corp. TRACE GC Ultra Gas Chromatographs) and a mass spectrometer detector (ISQ Single Quadrupole Mass Spectrometer, THERMO Scientific Corp., Waltham, MA, USA). A TG-5MS column (30 m × 0.25 mm i.d., 0.25 m film thickness) was installed in the GC-MS system, manufactured by THERMO Scientific. The studies were conducted using the following temperature program, employing a split ratio of 1:10 and a flow rate of 1.0 mL/min of helium as the carrier gas: After increasing from 60 °C for one minute at a pace of 3.0 °C per minute, the temperature was maintained at 240 °C for one minute. The injector and detector were kept at 240 degrees. 0.2 microliters of the mixtures were administered as diluted samples (1:10 hexane, *v*/*v*) in each of the two tests. Mass spectra were created using electron ionization (EI) at 70 eV with a spectral range of 40–450 *m*/*z*. Analytical mass spectra were used to identify the bulk of the substances as genuine substances in the Wiley spectral library.

### 2.3. Conditions for Experiment

The South Valley University institutional animal care and use committee authorized the experimental methodology, and the testing on the birds was conducted in accordance with institutional animal housing and handling guidelines (approval code: SVU-AGRI-3-2022). In the safe building, three-tier wire floor battery cages were used to rear chicks. In cages with bottoms made of iron slats, the chicks of each replicate were distributed. The length, width, and height of the cages were 120, 70, and 50 cm, respectively. The chicks had complete access to mash diets and water throughout the experiment. Throughout the trial period of 35 days, the average outdoor low and high temperatures were 34.9 and 40.2 °C (38.7 ± 2.6 °C), respectively, with relative humidity readings of 35.5 and 39.0% (37.25%). When the chickens were 1 to 7, 8 to 21, and 22 to 35 days old, the brooding temperatures (indoors) were 38.3, 35.2, and 29.5 °C, respectively, with relative humidity readings of 50.5 and 55.5% (52.25%). The birds were vaccinated with vaccines against Infectious Bronchitis viruses and Newcastle disease (at 7 and 16 days) and Gumboro (at 12 days).

### 2.4. Broiler Performance Parameters

The broiler body weight in each pen was recorded every week from the first day of the experiment until the end. Additionally, monitoring feed residue on the same days that the birds were weighed allowed for the estimation of feed consumption for each pen in between weighings. Feed per gain was used to calculate the feed conversion ratio by dividing the weight of feed consumed by the increase in body weight in each pen. Adjustments were made to feed intake and body weight for bird mortalities.

### 2.5. Carcass Criteria and Internal Organs

Birds were fasted overnight while having access to water when they were 35 days old. For each treatment, 32 selected birds (four birds/replicate that represented the pen) were weighed, slaughtered, and plucked. After removing the neck, head, shanks, viscera, digestive tract, spleen, liver, gizzard, heart, and belly fat, the remaining parts of the body were weighed to determine the dressed weight. The proportions of the dressing, breast, drumstick, and thigh to the live weight were calculated. The percentages of liver, empty gizzard, heart, and belly fat of broilers to live body weight were calculated.

### 2.6. Meat Quality Measurements

To measure cooking loss, pH, and water-holding capacity (WHC), each broiler chicken’s left drumstick, thigh, and side of the breast muscle were used. A pH-meter (Lenzkirch, Germany-based Knick Model 205, AG) was used to measure the pH values 24 h after death after homogenizing breast or leg muscles (1 g) with iodoacetate, according to Korkeala et al. [18]. To calculate the water holding capacity (WHC), one gram of breast or leg muscle was inserted inside a tube on tissue paper and centrifuged at 1500× *g* for four minutes. The amount of water that remained after centrifugation was determined by drying the samples at 70 °C for an overnight period (Nakamura and Katoh) [19]. The WHC was computed as follows: $$WHC=\frac{\mathrm{weight}\;\mathrm{after}\;\mathrm{centrifugation}-\mathrm{weight}\;\mathrm{after}\;\mathrm{drying}}{\mathrm{initial}\;\mathrm{weight}}\times 100$$

### 2.7. Blood Sample Collection and Analysis

At 35 days of age, blood was drawn from the wing veins of 32 birds per treatment (4 birds were used per replicate that represented the pen) and put in vacutainer tubes for serum collection. Before drawing blood, the feeder wasn’t taken out of the pens. The blood was centrifuged for 10 min at 3000× *g* at room temperature. The serum was collected in Eppendorf tubes and stored at −20 °C until analysis. The clinical chemistry analyzer SBA 733 plus (Sunostik Medical Technology Co., Ltd., Changchun, China) and kits (Diagnostic kits, Biodiagnostics, Cairo, Egypt) were used to perform a colorimetric analysis of triglycerides, aspartate aminotransferase, and alanine transaminase as a liver function test, as well as urea and creatinine as a kidney function test. 

### 2.8. Statistical Analysis

The experimental unit for all statistical analysis was the pen. Using the Shapiro-Wilks test, all data were examined for normal distribution (W > 0.05). Then, Duncan multiple range tests were employed to compare means after a one-way ANOVA was carried out using the SAS 9.2 [20] statistical program. The mean and standard error of the mean (SEM) were used to express values. *p* < 0.05 was used to declare significance. *p*-values were expressed as “0.001” rather than the real value when they were less than “<0.001”.

## 3. Results

### 3.1. Oregano and Peppermint Active Components

The data revealed that the main compounds of BLCO were: Carvacrol 85.08%, o-Cymene 3.78%, γ-Terpinene 3.19%, Thymol 1.32%, Thujene 0.84%, α-Pinene 0.64%, Camphene 0.19%, β-Pinene 0.12%, β-Myrcene 0.34%, α-Phellandrene 0.16%, α-Terpinene 0.62%, trans-Sabinene Hydrate 0.06%, cis-β-Terpineol Borneol 0.17%, 4-Terpineol 0.09%, trans-Caryophyllene 0.49%, β-Bisabolene 0.08%, Caryophyllene oxide 0.08%, and p-Cymene 0.06%. The main components of BLCP were: 38.12% L-menthol, 33.35% menthone, 8.35% isomenthone, 6.99% 1,8-Cineol, 2.4% pulegone, 1.6% linalool, 1.29% myrcene, 0.91% isomenthyl acetate, 0.53% α-terpinplene, 0.33% caryophyllen oxide, 0.30% neomenthyl acetate, 0.27% linalool oxide, and 0.22% linalool trans. 

### 3.2. Growth Performance

The impacts of BLCO, BLCP, and their combination on the broiler’s productive performance are described in Table 2 and Table 3. According to the findings, supplementing BLCO and BLCP individually as well as in combination significantly (*p* < 0.001) improved body weight during 21 and 35 days of age and daily body weight gain in comparison to the control group at 1 to 21, 22 to 35, and 1 to 35 days of age (Table 2). The BLCO and BLCP combination resulted in a higher body weight in comparison to other treatments at 35 days of age. Body weight and body weight gain did not differ between BLCO and BLCP. In comparison to the other groups, the BLCO and BLCP blends exhibited higher body weight gain. When BLCO and BLCP or their combination were added to broiler diets, improvements in daily feed conversion ratio were seen at 1–21, 22–35, and 1–35 days (Table 3). However, there was no difference in intake of feed (*p* > 0.05) between BLCO, BLCP, or their combination (Table 3). In comparison to other treatments, the combination of the BLCO and BLCP groups provided the best results in terms of body weight, weight gain, and feed conversion ratio.

### 3.3. Physicochemical Properties of Meat

In terms of the physicochemical characteristics of meat (Table 4), adding BLCO, BLCP, or both at 35 days of age under hot temperatures considerably reduced cooking and drip loss in the breast and leg of broilers in comparison to the control. Conversely, BLCO, BLCP, or their combination had no effect (*p* > 0.05) on the pH values in the breast and leg of broilers (Table 4). The combination of BLCO and BLCP groups did not show any significant difference in pH value, cooking, or drip loss in the breast and legs of broilers when compared to BLCO or BLCP alone.

### 3.4. Carcass Criteria and Organs

According to the data on internal organs and carcass parameters (Table 5 and Table 6), when BLCO, BLCP, or their combination were added to broiler chickens’ diet, the percentages of the dressing, breast, drumstick, and thigh significantly (*p* < 0.05) improved, but the percentage of abdominal fat decreased at 35 days of age (Table 5). Gizzard percentage increased (*p* < 0.05) when BLCO, BLCP, or their combination was supplemented compared to control (Table 6). On the other hand, BLCO, BLCP, or their combination had no effects (*p* > 0.05) on the broilers’ liver, heart, spleen, small intestine, or cecum (Table 6).

### 3.5. Serum Biochemicals 

The effects of BLCO, BLCP, and their combination on a variety of serum biochemical parameters in broilers are shown in Figure 1, Figure 2 and Figure 3. When BLCO, BLCP, or their combination were added to the diets, serum cholesterol and triglycerides decreased (*p* < 0.05) compared to the control (Figure 1). Moreover, serum cholesterol was lowest (*p* < 0.05) in combination diets in comparison to other treatments. The serum levels of ALT and AST were significantly lower in broilers fed BLCO, BLCP, or blend diets compared to control (Figure 2). In broilers fed BLCO, BLCP, or combination diets, the serum levels of creatinine and urea decreased (Figure 3), compared to the control. The serum ALT, AST, creatinine, and urea were lowest (*p* < 0.05) in the combination diet in comparison to other treatments.

## 4. Discussion

The efficacy of the use of various phytogenics in different studies is dependent on a variety of variables, including the plant species present, the dosage administered, the manner and frequency of application, the age of the bird, and environmental stressors, making it challenging to directly compare these studies [15]. Furthermore, it is still unclear whether some effects are the result of a synergistic interaction between several factors or a single factor [9,12]. Therefore, it is essential to determine the chemical composition of an oregano and peppermint to define its ideal composition, and discovering the chemical composition of BLCO and BLCP before usage may provide helpful information about its impact on birds. The composition of the BLCO and BLCP utilized in this investigation was consistent with that published in the literature [21,22,23]. Carvacrol and thymol constitute the majority of the oregano BLC’s, accounting for up to 85% of it in total [24]. The number of active components in BLCO and BLCP varied depending on the environmental factors affecting plant development and genetic variety, as well as processing and storage [21]. Carvacrol has antibacterial, anti-inflammatory, and antioxidant activities [25,26,27]. hese properties may improve the health and performance of broilers.

In this experiment, incorporating BLCO and BLCP either separately or in combination with broiler diets increased body weight, weight gain, and the ratio of feed to body gain in comparison to the control group throughout the starter, grower, or for the whole period. The medicinal plant’s active ingredients may be the reason broiler chicks fed BLCO and BLCP grew faster. According to Amad et al. [28], broilers fed phytogenics that contained the active substances in thyme essential oils and star anise had a better feed conversion ratio, which can contribute to higher economic efficiency in broiler production. Additionally, the BLCO and BLCP dramatically enhanced the feed conversion ratio and boosted body weight gain, which may be associated with an improvement in appetite and feed efficiency. Moreover, the increased broiler growth performance in birds fed BLCO and BLCP under hot environmental conditions due to these medicinal plants could be explained by the presence of carvacrol and menthol, respectively, in them. Broiler chicks’ appetites appear to be stimulated, and their feed utilization appears to be improved by the presence of active chemicals. The herbal active components have been shown to improve digestion and nutrient absorption and have antimicrobial and antioxidant properties [29]. These properties may have helped the broiler chicks increase their growth. Likewise, herbal BLC appears to have an effective impact on enhancing daily body weight gain by lowering the effects of gastrointestinal problems [30,31]. This may strengthen the digestive system and increase feed efficiency. According to Toghyani et al. [32] findings, broiler chicks fed diets containing peppermint leaves had better early-life growth performance. In a similar vein, Nobakht et al. [33] reported that supplementing dried *Mentha pulegium* at a rate of 5 g/kg enhanced broiler growth performance at 42 days of age. In comparison to the control group, broilers given 1.5% peppermint leaves significantly increased body weight, weight gain, and the ratio of feed to body gain [34]. Additionally, Mimica-Dukic et al. [35] found that the pharmacological characteristics of mint oil enhanced hepatic antioxidant status, increased bile production, and enhanced growth performance. Due mostly to its active ingredients, oregano (*Origanum vulgare*) is a common medicinal herb noted for its antioxidant, antibacterial, immunity enhancement, and hypocholesteremic features [15,36]. Furthermore, the growth performance, gastrointestinal secretions, metabolism, and health of broilers were all found to be improved by oregano BLC [14].

The experimental results in the current study showed that BLCO and BLCP, alone or in combination, effectively reduced the drip loss and cooking loss percentage of the breast and leg muscles at 35 days of age. These findings showed that the BLC of oregano and peppermint supplementations in broilers had the ability to reduce breast meat cooking loss and consequently avoid tissue damage and cooking loss of muscle tissues. These reductions in cook loss may be attributable to phytogenic substances that have antioxidant activity, may protect chicken meat, and may also increase the oxidative stability of meat [31,37]. 

For studies related to carcass parameters, BLCO and BLCP improved dressing, breast, leg meat, and abdominal fat percentages and showed no negative effects on internal organs when compared to the control in hot weather conditions. The reduction in abdominal fat may be related to phytogenics in the diet of broiler chickens that improved the activities of trypsin and amylase [38,39] and bile acid secretion [28]. In the current study, supplementation with BLCO, BLCP, or their combination significantly decreased serum cholesterol and triglycerides compared to control. AST, ALT, creatinine, and urea blood levels were decreased in broilers fed BLCO, BLCP, or combination diets. The results are in line with the usual physiological range for chickens. Serum ALT, AST, urea, and creatinine are generally regarded as indicators of liver disease and damage to the kidneys, respectively. The results from the current study may suggest that BLCO and BLCP are safe for poultry and that feeding these substances to broilers has no adverse effects. According to Reis et al. [40], broilers receiving a phytogenic additive containing carvacrol, thymol, and cinnamaldehyde showed no alterations in their AST or ALT activity. The results of Abdel-Ghany and El-Metwally [41], who indicated that the antioxidant content of marjoram BLC could prevent liver and kidney damage, are similar to our results. According to Omer et al. [42], adding oregano leaves and fennel seeds to the diets of rabbits resulted in a substantial decrease in total cholesterol. These results led to the conclusion that diets containing BLCO and BLCP were appropriate for improving productive performance and meat quality. Due to the dearth of information on the supplementation of BLCO and BLCP in broilers, this might be a unique nutritional way to enhance growth performance, meat quality, health status, and reduce abdominal fat in hot environmental conditions.

## 5. Conclusions

Oregano and peppermint’s bioactive lipid components might be utilized as efficient novel nutritional compounds to enhance broiler performance, carcass criteria, meat quality, and liver and kidney functions, in addition to lowering serum cholesterol. The broiler chickens that received the combination of BLCO and BLCP showed the best performance and meat quality. More thorough research is still required to clarify the mechanism of action and optimum concentrations of BLCO and BLCP on the physiological and nutritional responses of broilers in various circumstances.

## Figures and Tables

**Figure 1 vetsci-10-00612-f001:**
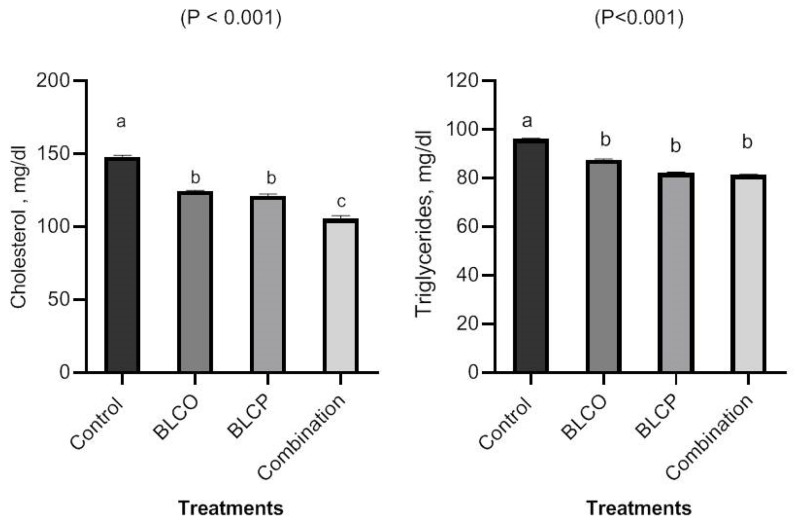
Serum cholesterol and triglycerides of broilers in response to control, bioactive lipid compounds of oregano (BLCO), peppermint (BLCP), and their combination at 35 days of age. ^a–c^ The bars in figures with different superscripts are different (*p* ˂ 0.05).

**Figure 2 vetsci-10-00612-f002:**
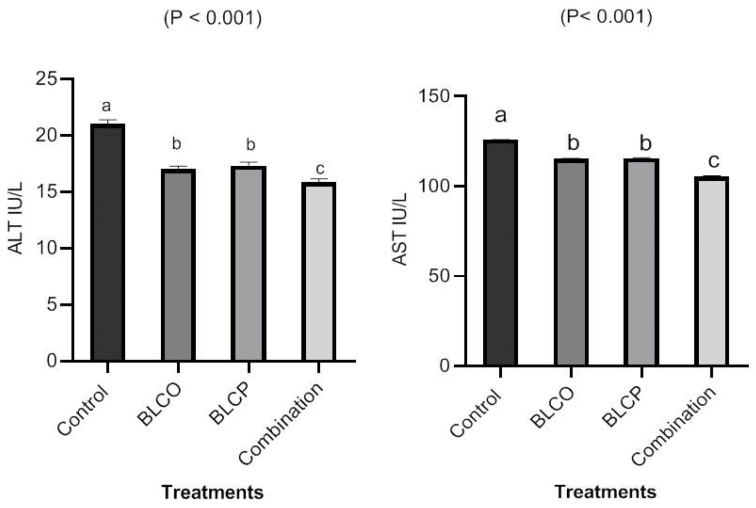
Serum ALT and AST as liver functions of broilers in response to control, bioactive lipid compounds of oregano (BLCO), peppermint (BLCP), and their combination at 35 days of age. ^a–c^ The bars in figures with different superscripts are different (*p* ˂ 0.05).

**Figure 3 vetsci-10-00612-f003:**
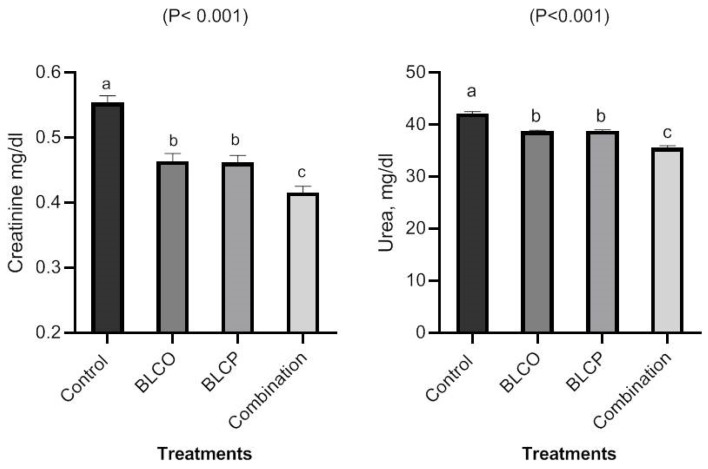
Serum urea and creatinine as kidney functions of broilers in response to control, bioactive lipid compounds of oregano (BLCO) peppermint (BLCP), and their combination at 35 days of age. ^a–c^ The bars in figures with different superscripts are different (*p* ˂ 0.05).

**Table 1 vetsci-10-00612-t001:** Chemical composition of the standard diet (as-fed basis).

Ingredients, g/kg	Starter Diet	Grower Diet
Corn	276	300
Sorghum	276	300
Soybean (44% CP)	285	250
Corn gluten (60% CP)	95.0	60.0
Vit and Min. Premix ^a^	3.00	3.00
Oil of Sunflower	30.0	55.2
Dicalcium phosphate	20.0	18.0
Limestone	10.0	10.00
Salt	3.80	3.80
DL-methionine	0.40	---
L-lysine HCl	1.00	---
Total	1000	1000
Analysis of chemical composition, g/kg		
Dry matter	925	924
Crude protein	233	216
Ether extract	53.7	57.5
Crude fibre	25.8	37.8
Ash	67.4	61.8
Ca	13.22	12.84
P	7.05	7.21
GE, MJ/kg	18.55	19.18

^a^ Supplied per kg diet, vitamin A (1900 IU), D3 (1300 IU), E (10,000 mg), B1 (1000 mg), K3 (1000 mg), B2 (5000 mg), B6 (1500 mg), and B12 (0.046 mg), BHT (10,000 mg), along with biotin (50 mg), pantothenic acid (10,000 mg), folic acid (1000 mg), and nicotinic acid (30,000 mg). Contains Mn (60 mg), Fe (30 mg), Zn (50 mg), Se (0.1 mg), Cu (4 mg), I (3 mg), and Co (0.1 mg).

**Table 2 vetsci-10-00612-t002:** Effects of bioactive lipid compounds of oregano (BLCO) and peppermint (BLCP) and their combination on body weight and body weight gain of broiler chickens.

Items	Body Weight, g			Daily Body Weight Gain, g		
	1 Day	21 Days	35 Days	1–21 Days	21–35 Days	1–35 Days
Control	41.78	868 ^c^	1988 ^c^	41.30 ^b^	79.96 ^c^	57.22 ^c^
BLCO	41.43	967 ^a^	2158 ^b^	46.80 ^a^	85.77 ^ab^	62.25 ^ab^
BLCP	41.57	976 ^a^	2143 ^b^	46.71 ^a^	83.38 ^b^	61.81 ^b^
Combination	41.63	989 ^a^	2209 ^a^	47.36 ^a^	87.15 ^a^	63.75 ^a^
SEM	0.45	8.79	17.03	0.44	0.75	0.50
*p*-Value	0.478	0.001	0.001	0.001	0.002	0.001

^a–c^ The values in each column with different superscripts are different (*p* ˂ 0.05). BLCO: bioactive lipids compounds of oregano. BLCP: bioactive lipids compounds of peppermint. SEM: Standard Error of Means (*n* = 8).

**Table 3 vetsci-10-00612-t003:** Effects of bioactive lipid compounds of oregano (BLCO) and peppermint (BLCP) and their combination on feed conversion ratio and feed intake of broiler chickens.

Items	Daily Feed Intake, g			Daily Feed Conversion Ratio		
	1–21 Days	21–35 Days	1–35 Days	1–21 Days	21–35 Days	1–35 Days
Control	59.46	146.1	101.1	1.440 ^a^	1.827 ^a^	1.768 ^a^
BLCO	60.48	145.5	101.5	1.321 ^b^	1.697 ^b^	1.631 ^b^
BLCP	59.58	148.2	102.2	1.276 ^c^	1.781 ^ab^	1.654 ^b^
Combination	60.34	149.7	103.3	1.274 ^c^	1.698 ^b^	1.622 ^b^
SEM	0.15	0.76	0.32	0.010	0.017	0.013
*p*-Value	0.052	0.163	0.079	0.001	0.017	0.001

^a–c^ The values in each column with different superscripts are different (*p* ˂ 0.05). BLCO: bioactive lipids compounds of oregano. BLCP: bioactive lipids compounds of peppermint. SEM: Standard Error of Means (*n* = 8).

**Table 4 vetsci-10-00612-t004:** Effects of bioactive lipid compounds of oregano (BLCO) and peppermint (BLCP) and their combination on the physicochemical meat quality of broiler chickens.

Items	Breast Muscle			Leg Muscle		
	pH Value	Drip Loss, %	Cook Loss, %	pH Value	Drip Loss, %	Cook Loss, %
Control	5.92	22.71 ^a^	22.48 ^a^	5.90	22.16 ^a^	22.45 ^a^
BLCO	5.89	19.45 ^b^	19.21 ^b^	5.93	19.45 ^b^	19.30 ^b^
BLCP	5.91	19.39 ^b^	19.24 ^b^	5.91	19.26 ^b^	19.26 ^b^
Combination	5.90	19.22 ^b^	19.12 ^b^	5.92	19.32 ^b^	19.24 ^b^
SEM	0.03	0.27	0.26	0.01	0.23	0.26
*p*-Value	0.364	<0.001	<0.001	0.112	<0.001	<0.001

^a,b^ The values in each column with different superscripts are different (*p* ˂ 0.05). BLCO: bioactive lipids compounds of oregano. BLCP: bioactive lipids compounds of peppermint. SEM: Standard Error of Means (*n* = 8).

**Table 5 vetsci-10-00612-t005:** Effects of bioactive lipid compounds of oregano (BLCO) and peppermint (BLCP) and their combination on carcass parameters of broiler chickens.

Items	Carcass Characteristics, %				
	Dressing	Breast	Thigh	Drumstick	Abdominal Fat
Control	69.41 ^c^	20.45 ^b^	11.37 ^b^	9.84 ^b^	1.199 ^a^
BLCO	72.25 ^b^	22.98 ^a^	11.99 ^a^	10.39 ^a^	0.848 ^b^
BLCP	72.79 ^a b^	23.15 ^a^	12.05 ^a^	10.39 ^a^	0.744 ^c^
Combination	73.69 ^a^	23.70 ^a^	12.12 ^a^	10.51 ^a^	0.700 ^c^
SEM	0.341	0.280	0.101	0.090	0.039
*p*-Value	<0.001	<0.001	0.015	0.016	<0.001

^a–c^ The values in each column with different superscripts are different (*p* ˂ 0.05). BLCO: bioactive lipids compounds of oregano. BLCP: bioactive lipids compounds of peppermint. SEM: Standard Error of Means (*n* = 8).

**Table 6 vetsci-10-00612-t006:** Effects of bioactive lipid compounds of oregano (BLCO) and peppermint (BLCP) and their combination on the internal organs of broilers.

Items	Internal Organs, %						
	Liver	Heart	Gizzard	Spleen	Pancreas	Small Intestine	Cecum
Control	1.962	0.538	1.361 ^c^	0.093	0.016	2.830	0.626
BLCO	1.966	0.525	1.394 ^b^	0.094	0.017	2.891	0.647
BLCP	1.977	0.529	1.395 ^b^	0.092	0.010	2.848	0.644
Combination	1.85	0.553	1.421 ^a^	0.093	0.015	2.860	0.646
SEM	0.007	0.004	0.006	0.002	0.006	0.009	0.005
*p*-Value	0.602	0.129	0.004	0.557	0.421	0.114	0.324

^a–c^ The values in each column with different superscripts are different (*p* ˂ 0.05). BLCO: bioactive lipids compounds of oregano. BLCP: bioactive lipids compounds of peppermint. SEM: Standard Error of Means (*n* = 8).

## Data Availability

The datasets generated and/or analyzed during the current study are available from the corresponding author on reasonable request.

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
