# Peer review of "Bioactive Lipid Compounds as Eco-Friendly Agents in the Diets of Broiler Chicks for Sustainable Production and Health Status"

_vetsci, 2023, doi:10.3390/vetsci10100612_

Round 1
Reviewer 1 Report
The manuscript is very interesting, well organized and I recommend it publication after addressing the following commments:
Line 26: Please delete (BW)
The introduction needs to be updated with recent references.
Line 42: please add the following paragraph to highlight the most negative impacts of heat stress on poultry. Heat stress can also lead to disruption of gut tight junctions, reactive oxygen species (ROS) production that increases intestinal permeability, endotoxemia and systemic inflammation, hepatic inflammation, reduction of calcium absorption and decreasing the metabolism of vitamin D3 (cholecalciferol) to 1,25-dihydroxy vitamin D3 (1,25(OH)2D3) as well as growth retardation (Doi: https://doi.org/10.51585/gjvr.2023.1.0051)
Line 49: Please add additional recent references: Doi:https://doi.org/10.51585/gjvr.2021.3.0018
Line 57: Please add recent references https://doi.org/10.3390/vetsci10010055
Line 85: Aviagen 2019: Is this a refernce?
Line 88: Clevenger-type apparatus (please add the supplier, City, and Country); please revise the source of all materials used in this study
Line 105: Please provide more data about the vaccination program against the used chickens.
Line 168: The title should include Organo also!
Why do the authors not analyze the modulation of gastrointestinal microbiota?
Line 308: Mentha pulegium should be in italic
Minor editing of English language required
Reviewer 2 Report
The paper would be benefited from addressing or reconsidering to modify the following:
1/ The Abstract is not written in the correct format. For example, the subtitle ‘simple summary is not necessary, and it didn’t follow journal guideline.
2/ The Abstract missed a brief introduction about the topic of interest
3/ Line 38, reference writing. if you start according to X (Author name), write the year and then [the reference number]. Apply this in all part of the paper.
4/ in section 2.3 ‘Experimental Conditions’, IACUC number should be provided.
5/Line 128, is it overnight starvation or fasting? Use the proper term.
6/ Section 2.7 ‘Blood sampling and laboratory analyses’ is copied from somewhere. For example, ‘how many birds per replicate, please mention’ and protocol/reference for the blood collection should be provided.
7/In the result section, Figure 1-3, the statistical comparison is not shown. Indicate on the figure itself. Only P-Value seems showed.
